# Monoclonal Antibodies for the Treatment of Multiple Myeloma: An Update

**DOI:** 10.3390/ijms19123924

**Published:** 2018-12-07

**Authors:** Hanley N. Abramson

**Affiliations:** Department of Pharmaceutical Sciences, Wayne State University, Detroit, MI 48202, USA; ac2531@wayne.edu

**Keywords:** myeloma, daratumumab, elotuzumab, isatuximab, CD38, JNJ-63723283, denosumab, checkpoint inhibitors, BCMA, bispecific T-cell engager, antibody-drug conjugates

## Abstract

The past two decades have seen a revolution in multiple myeloma (MM) therapy with the introduction of several small molecules, mostly orally effective, whose mechanisms are based on proteasome inhibition, histone deacetylase (HDAC) blockade, and immunomodulation. Immunotherapeutic approaches to MM treatment using monoclonal antibodies (mAbs), while long in development, began to reap success with the identification of CD38 and SLAMF7 as suitable targets for development, culminating in the 2015 Food and Drug Administration (FDA) approval of daratumumab and elotuzumab, respectively. This review highlights additional mAbs now in the developmental pipeline. Isatuximab, another anti-CD38 mAb, currently is under study in four phase III trials and may offer certain advantages over daratumumab. Several antibody-drug conjugates (ADCs) in the early stages of development are described, including JNJ-63723283, which has attained FDA breakthrough status for MM. Other mAbs described in this review include denosumab, recently approved for myeloma-associated bone loss, and checkpoint inhibitors, although the future status of the latter combined with immunomodulators has been clouded by unacceptably high death rates that caused the FDA to issue clinical holds on several of these trials. Also highlighted are the therapies based on the B Cell Maturation Antigen (BCMA), another very promising target for anti-myeloma development.

## 1. Introduction

Multiple myeloma (MM) is the second most common hematological cancer, accounting for about 10% of all blood cancers. It is estimated that approximately 30,770 (53.3% male) new cases are diagnosed in the US annually with an annual death rate of 12,770, representing about 2.9% of all cancer deaths [1]. The disease is typified by clonal expansion of transformed plasma cells in the bone marrow and is associated with the tetrad of elevated plasma calcium, renal failure, anemia, and bone lesions (CRAB). Also accompanying the disease is the presence in the blood or urine of paraproteins of diagnostic significance: immunoglobulin free light chains, which contribute to kidney damage and amyloidosis, monoclonal protein (M protein or M spike), and β-2 microglobulin, a major indicator of patient survival. Two asymptomatic pre-malignant stages, known as monoclonal gammopathy of undetermined significance (MGUS) and smoldering myeloma constitute part of a continuum leading up to clinical disease.

Although still considered incurable, recent advances in the treatment of MM have begun to change the picture, resulting in a near doubling of the five-year survival rate to around 50% since the late 1980s. The first major therapeutic advance in MM treatment was the demonstration of the efficacy of the alkylating agent melphalan, especially when combined with low-dose corticosteroids, such as dexamethasone or prednisone. To this was added autologous stem cell transplantation (ASCT) in the 1990s. Over the past 20 years, this approach to myeloma therapy has shifted dramatically with the identification of a number of molecular targets relevant to the disease and the development of small molecular weight (less than 900 kDa) agents, mostly orally effective, impacting those targets. These new classes of anti-myeloma drugs include immunomodulators (thalidomide, lenalidomide, and pomalidomide), proteasome inhibitors (bortezomib, carfilzomib, and ixazomib), and histone deacetylase (HDAC) blockers (panobinostat) [2]. Although initial treatment of the disease employing one or more of these drug classes, either as monotherapy or, more often, in combination, frequently induces remission, patients very often experience relapse or resistance. As a possible means of overcoming remaining barriers to disease cure, attention has been drawn increasingly to immunotherapeutic measures with the potential for improved targeting of myeloma cells [3]. Indeed, these efforts already have begun to bear fruit with the FDA approval in 2015 of two monoclonal antibodies (mAbs) for the disease, daratumumab and elotuzumab [4]. The present review is intended to provide a comprehensive overview of the current state of the pipeline leading to the introduction of new mAbs for the treatment of MM. The antibodies noted in this review were selected from information contained in 2501 myeloma-based trials in the clinicaltrials.gov database that were initiated between 1 November 1999 and 8 June 2018, as well as a review of the pertinent reported data contained in the PubMed database and meeting abstracts. The discussion that follows is divided primarily on the basis of the putative molecular and/or cellular target(s) for each mAb. 

## 2. CD38

The cell surface single-chain transmembrane glycoprotein CD38, which is expressed at high levels in both normal and malignant plasma cells, serves as a counter receptor for CD31, as well as an ectoenzyme with cyclic ADP ribose hydrolase activity. Although they exhibit relative differences, CD38 antibodies cause apoptosis of myeloma cells through a variety of mechanisms, primarily resulting from Fc-dependent processes that include antibody-dependent cellular cytotoxicity (ADCC), antibody-dependent cellular phagocytosis (ADCP), and complement-dependent cytotoxicity (CDC). Moreover, crosslinks formed between CD38-bound antibody on MM cells and Fcγ receptors on effector cells may play a key role in direct cytotoxicity of myeloma cells by inducing apoptosis through disruption of intracellular signaling cascades [5]. Finally, anti-CD38 antibodies possess immunomodulatory effects that block both regulatory T- and B-cells, as well as myeloid-derived suppressor cells (MDSCs) [6]. 

Daratumumab, which targets CD38^+^ cells, received FDA breakthrough status for MM in May 2013. This fully humanized IgG1κ mAb subsequently was approved in November 2015 for patients whose disease had failed at least three prior therapies including an immunomodulator and a proteasome inhibitor [7]. Approval was granted following the results of the multicenter phase II SIRIUS trial (NCT01985126) [8], the phase III POLLUX trial (NCT02076009) [9], and the phase III CASTOR study (NCT02136134) [10]. The latter two demonstrated beneficial results upon addition of daratumumab to lenalidomide/dexamethasone or bortezomib/dexamethasone regimens, respectively. The recognition of daratumumab as a highly efficacious component of anti-myeloma regimens in relapsed/refractory MM (RRMM) is supported by five recently published network meta-analyses (NMAs) of randomized controlled trials (RCTs). Botta et al. [11] in a study of 18 trials concluded that daratumumab + lenalidomide + dexamethasone ranked first among other regimens in terms of activity, efficacy, and tolerability. Similar conclusions were reached by Dimopoulos et al. [12] from analysis of 82 RCTs, Zhang et al. (13 RCTs) [13], as well as by van Beurden-Tan et al. [14], who in their study of 17 RCTs found that the combination lowered the risk of disease progression or death by 87% compared to 81% for bortezomib + dexamethasone and 63% for lenalidomide + dexamethasone. Finally, Maiese, et al. [15] published a meta-analysis of 27 RCTs in which they concluded that combinations of daratumumab/dexamethasone with either bortezomib or lenalidomide were comparatively efficacious in terms of both significantly higher objective response rates (ORRs) and improved progression-free survival (PFS) in heavily pre-treated RRMM subjects while the drug’s safety profile was very favorable. Moreover, trials of daratumumab as monotherapy in RRMM also have demonstrated the drug’s efficacy at doses of 16 mg/kg. with reported ORRs and PFS of 36% and 6 months (NCT00574288) [16] and 29% and 3.7 months [8]. The most frequently observed hematologic grade 3 or 4 adverse events in the daratumumab-based trials were neutropenia and thrombocytopenia. Grade 3 or 4 Infusion-related reactions were seen in only 5% of patients. Data also is starting to accumulate supporting daratumumab combinations as frontline therapy for newly diagnosed MM. For example, one phase III study (NCT02195479; ACYONE) of 706 transplant-ineligible patients showed that addition of daratumumab to a bortezomib, melphalan, and prednisone regimen resulted in a significantly lower risk of disease progression or death, although patients in the daratumumab arm tended to experience more grade 3 or 4 infections [17]. In addition, recent reports from the CENTAURUS (NCT02316106) [18] and AQUILA (NCT03301220) [19] trials have demonstrated a potential role for daratumumab in high-risk smoldering MM. A select number of ongoing daratumumab trials in MM are listed in Table 1.

Isatuximab (SAR650984), a chimeric mouse/human CD38-targeting IgG1κ mAb, has shown some significant differences from daratumumab in terms of its mechanism of action in that its antitumor efficacy appears to be primarily dependent on ADCC with less contribution from CDC (75677) [6]. Also, in contrast to daratumumab, crosslinking induced by isatuximab is not a prerequisite for initiation of apoptosis [20,21]. In addition, although of unknown clinical significance, compared to other anti-CD38 agents isatuximab is much more potent as an inhibitor of ectoenzyme activity. Preclinical studies revealing that the cell-killing ability of isatuximab was enhanced when combined with an immunomodulator [20], informed a phase Ib study (NCT02283775) in which a combination of isatuximab with pomalidomide/dexamethasone elicited at least partial responses with manageable toxicity in 62% of 26 RRMM patients [22,23]. Another phase Ib study (NCT01749969) of 57 heavily pretreated RRMM patients with lenalidomide/dexamethasone showed the combination to be well tolerated with a median PFS of 8.5 months [24]. A large-scale (300 patients) multinational study (NCT02990338) recently was initiated comparing pomalidomide/dexamethasone with and without isatuximab [25]. Moreover, there currently are three other ongoing large-scale phase III trials aimed at comparing the efficacy of adding isatuximab to standard anti-proteasome- and/or immunomodulator-based myeloma regimens (see Table 2).

Another anti-CD38 mAb in development is the fully humanized IgG1λ MOR03087 (MOR-202), which has been shown to be well tolerated at weekly doses up to 16 mg/kg. in a phase I/IIA trial in RRMM patients both as a single agent and in combination with immunomodulators (NCT01421186) [32,33]. In addition, Takeda Oncology has developed two anti-CD38 mAbs that currently are in the early stages of clinical development for RRMM: TAK-573 (NCT03215030), in which the IgG4 antibody is conjugated to an attenuated form of interferon α, and the fully humanized IgG1λ TAK-079 (NCT03439280 and NCT02219256) [34].

## 3. SLAM Family Proteins

The signaling lymphocytic activation molecule family (SLAMF) of surface proteins is comprised of nine members, four of which are highly expressed on plasma cells from MM patients regardless of disease stage: SLAMF2 (CD48), SLAMF3 (CD229; Ly9), SLAMF6 (CD352), and SLAMF7 (CS1 or CD319). Of these, only SLAMF3 has yet to be included in target-directed clinical trials for MM [35]. In November 2015 elotuzumab (HuLuc63), a humanized mAb that specifically binds to SLAMF7, received FDA approval for use in combination with lenalidomide and dexamethasone in MM patients who had received one to three previous drugs. Approval was based on the results of the phase III ELOQUENT-2 trial (NCT01239797) involving 646 RRMM patients randomly assigned to receive lenalidomide and dexamethasone with or without the mAb. The group receiving elotuzumab demonstrated a reduced risk of disease progression (PFS: 19.4 vs. 14.9 months) and improved responses (ORR at one year: 68% vs. 57%; 41% vs. 27% at two years) compared to the control group (50953) [36]. Recently published data from the Phase II ELOQUENT-3 trial (NCT02654132) reported gains in both PFS (10.3 vs. 4.7 months) and ORR (53% vs. 26%) when elotuzumab was added to a pomalidomide/dexamethasone regimen in 117 RRMM patients [37]. NMA has been used as a means to generate data comparing efficacies of elotuzumab-containing regimens using results obtained from different trials. One example is Botta’s NMA, referred to earlier, in which it was found that regimens containing elotuzumab and dexamethasone + an immunomodulator ranked fifth among 18 RCTs in terms of PFS, ORR, and OS (overall survival) [11]. Meta-analysis of two RCTs that compared triplet elotuzumab-containing regimens with non-elotuzumab-based combinations concluded that three-drug regimens containing the mAb were more effective in terms of PFS and ORR [13]. In their NMA of results from the ELOQUENT-2 (elotuzumab + lenalidomide + dexamethasone) and POLLUX (daratumumab + lenalidomide + dexamethasone) trials cited earlier, Dimopoulos [12] calculated hazard ratios (HR) of 0.54 for PFS and 0.82 for OS, indicating that the POLLUX regimen is statistically the more efficacious of the two. This was similar to the HR value of 0.59 computed for PFS by Maiese [15] in their comparison of the two trials. A review of the clinicaltrials.gov website (accessed on 2 August 2018) reveals a total of 45 myeloma-based elotuzumab studies, either completed or in progress. A select number of those trials are included in Table 3. Details concerning both preclinical and clinical studies of elotuzumab may be found in a number of extensive recently published reviews [38,39,40,41,42].

Azintuxizumab vedotin (ABBV-838) is an antibody-drug conjugate (ADC) in which a SLAMF7-targeted mAb is linked to the potent antimitotic compound monomethyl auristatin E (MMAE) via a cathepsin B-cleavable peptide linker [43,44]. Two phase I studies in RRMM patients were initiated with this conjugate. In one (NCT02951117), the ADC was combined with the BCl2 inhibitor venetoclax and dexamethasone while the other (NCT02462525) was intended to study the conjugate alone and in combination with pomalidomide/dexamethasone. However, both trials now have been terminated for unspecified reasons.

SGN-CD48A [52], an ADC in which a CD48-directed antibody is coupled to MMAE via a PEGylated β-glucosidase-cleavable linker and a stabilizing maleimide, is currently being evaluated in a phase I trial (NCT03379584) in RRMM patients. SLAMF6 is the target of SGN-352A, an ADC that is the focus of a recently initiated phase I trial for RRMM (NCT02954796), a study based in part on the observation that CD352 was found on 87% (13/15) of MM patient samples [53]. SGN-352A, one of a number of cysteine-engineered site-specific antitumor ADCs developed by Seattle Genetics, is constructed using an anti-CD352 antibody linked to a DNA minor groove binder pyrrolobenzodiazepine dimer [54] that, following internalization, becomes activated to cross-link DNA. 

## 4. Other Surface Antigens

### 4.1. ICAM-1 (CD54)

Intercellular Adhesion Molecule 1 (ICAM-1 or CD54), a surface receptor known to be upregulated in MM, is associated with poor prognosis and resistance to chemotherapy [55,56,57,58,59]. BI-505, a fully human immunoglobulin G1 directed against ICAM-1, was advanced to clinical trials for MM as the result of favorable preclinical studies [60]. Although well-tolerated in a phase I trial [61], a phase II study (NCT01838369) of BI-505 in smoldering myeloma patients failed to demonstrate evidence of clinical efficacy [62]. Furthermore, the FDA ordered the study halted when “an adverse cardio pulmonary event” was reported (https://www.ashclinicalnews.org/online-exclusives/clinical-trial-of-investigational-multiple-myeloma-drug-bi-505-placed-on-hold/ (accessed on 25 September 2018)).

### 4.2. CD40

CD40 is a costimulatory transmembrane protein belonging to the tumor necrosis factor (TNF) receptor family that is expressed by antigen-presenting cells. CD154, also a member of the TNF family, is the natural ligand for CD40 and is expressed on activated CD4^+^ T-cells [63]. Lucatumumab (HCD122; CHIR-12.12), a fully human anti-CD40 mAb, was shown to cause lysis of CD40^+^ (but not CD40^-^) myeloma cells via ADCC [64], leading to a phase I trial as monotherapy in RRMM patients (NCT00231166). Although the drug was well-tolerated, clinical activity against the disease was reported as only modest [65]. Dacetuzumab (SGN-40) [66] is another humanized mAb targeting CD40, but in this case with partial agonistic properties [67], that also produced ADCC-associated lysis of myeloma cells [68]. A trial of dacetuzumab combined with lenalidomide and dexamethasone (NCT00525447) in RRMM patients exhibited a modest 39% ORR [69] leading to termination of this drug’s further development. Moreover, it has been reported that no myeloma-based trials of lucatumumab are planned [70].

### 4.3. Fas

Fas (first apoptosis signal; CD95/APO-1) is a member of the TNF receptor family of cell surface proteins that induces caspase-dependent apoptosis in several types of cells, including tumor cells. Apoptotic signaling is triggered following binding of Fas to its ligand FasL, a transmembrane protein, also belonging to the TNF family, which is expressed on the surface of cytotoxic T cells among other cell types. Downregulation of Fas (or reduced responsiveness to FasL binding) in tumor cells, including myeloma cells, is a mechanism by which such cells evade recognition by the immune system [71]. APO010, a Fas receptor agonist that mimics the effects of FasL to induce apoptosis in MM cells, currently is the subject of a phase I trial in RRMM patients (NCT03196947).

### 4.4. FGFR3

A number of signal transduction pathways, including Ras-MAPK, PI3K-Akt, and phospholipase Cγ, are mediated by the fibroblast growth factor (FGF) family of proteins These transmembrane receptors exist as four different isoforms (FGFR1-4), each containing an immunoglobulin-like extracellular component linked to a cytoplasmic tyrosine kinase domain [72]. Overexpression of FGFR3 in particular has been implicated in cell proliferation, survival, invasion, angiogenesis, and drug resistance in a number of different cancers, including MM [73]. Especially important is the t(4;14) translocation of FGFR3, which is cited as one of the most frequently mutated genes in MM, being found in approximately 15% of MM patients and associated with poor prognosis and chemotherapeutic resistance [74]. MFGR1877S is an IgG1 anti-FGFR3 mAb that has exhibited strong activity in mouse xenograft models of t(4;14)-positive MM [75]. A phase I study (NCT01122875) of MFGR1877S in t(4;14)-positive RRMM has been completed although no results have been published.

## 5. Antibodies Targeting the Bone Marrow Microenvironment (BMM)

The extremely complex cross-talk between myeloma cells and the surrounding BMM plays a key role in sustaining tumor growth and in promoting many deleterious aspects of the disease, such as bone loss and drug resistance [76,77]. The microenvironment, which consists of T and B lymphocytes, natural killer (NK) cells, osteoclasts, osteoblasts, bone marrow stromal cells (BMSC), fibroblasts, endothelial cells, blood vessels, as well as the extracellular matrix (ECM), and a host of stroma-secreted growth factors, chemokines, and cytokines, can both impact myeloma cell growth and proliferation and in turn be affected by the tumor cells. For example, myeloma cells may promote growth of bone-forming osteoblasts while simultaneously suppressing bone-resorbing osteoclasts [76]. Understanding some of these relationships has enabled identification of several targets for mAb development.

### 5.1. IGF-1R

In addition to its production by the liver in response to circulating growth hormone [78], insulin-like growth factor-1 (IGF-1) also is produced by a number of extra-hepatic tissues, including the endothelial cells, osteoblasts, and BMSC in the BMM [79]. In the latter context, IGF-1 serves as an important growth stimulus for myeloma cells through its actions on the IGF-1 receptor (CD221) [80], a tyrosine kinase-linked activator of the PI3K/Akt pathway, which serves as a primary regulator of myeloma cell proliferation and apoptosis. The actions of IGF-1 on myeloma cells appear to be synergistic with those of IL-6 [81] although mechanisms independent of IL-6 also have been demonstrated [82]. The IGF-1 receptor, overexpression of which is associated with poor prognosis in MM [83,84,85], has been a major target of anti-myeloma mAb research and development [86], albeit with generally disappointing results. For example, in vitro work [87] with human myeloma cells that showed synergy between bortezomib and AVE1642, a humanized anti-IGF-1R mAb, led to a phase I trial (NCT01233895) of AVE1642 alone and with bortezomib in advanced myeloma. However, the response rates in both parts of the study were insufficient to justify further work with AVE1642 as a potential agent for MM [88]. Dalotuzumab (MK-0646) [89], another humanized anti-IGF-1R mAb is the subject of an ongoing phase I trial (NCT00701103) in solid tumors and MM. Although no data are available concerning efficacy or toxicity in myeloma, dalotuzumab has been reported as safe but dose-limiting hematologic toxicity, primarily grade 3 thrombocytopenia, has been observed in some patients [90]. Figitumumab (CP-751871), an IgG2-based mAb, has been the subject of a phase I study (NCT01536145) in RRMM patients but inadequate results have led to discontinuance of this agent for further consideration in MM [91].

### 5.2. IL-6

Monoclonal antibodies targeting the cytokine interleukin-6 (IL-6) have served as the basis for new drug development in a number of therapeutic areas, including MM. Secretion of IL-6 by the BMSC plays an important role in MM, promoting the growth and survival of malignant plasma cells through activation of a number of crucial pathways, such as JAK/STAT3, Ras/MAPK, and PI3K/Akt [80,92]. Elevated levels of serum IL-6 are associated with poor prognosis in the disease [93], as well as with protection against apoptosis induced by cytotoxic drugs such as dexamethasone [94]. Typically, serum levels of C-reactive protein (CRP), a surrogate marker of IL-6 inhibition, are reduced during anti-IL-6 therapy [95,96,97]. This inverse correlation between IL-6/CRP production and clinical response often serves as a useful parameter for evaluating dosing strategies when employing anti-IL-6 treatments.

Siltuximab (CNTO 328), a chimeric anti-IL-6 mAb, which has been approved by the FDA for multicentric Castelman’s disease [98], also has been the subject of several studies relating to MM. Although in vitro work suggested that siltuximab increases the anti-myeloma activity of bortezomib and dexamethasone [99,100], a phase II study (NCT00401843) of the combination of siltuximab and bortezomib in RRMM patients gave somewhat inferior results compared with bortezomib alone in terms of OS (30.8 vs. 36.8 months) and infection rates (62% vs. 49%) [101]. A phase II multi-center trial (NCT 00402181) of siltuximab and dexamethasone to establish the efficacy of anti-IL-6 therapy found that, although serum levels of CRP were reduced in almost all study subjects, complete responses were achieved in only 23% of the RRMM patients studied while an infection rate of 57% was noted [102]. A phase III trial (NCT01266811) designed to evaluate siltuximab or placebo in combination with bortezomib and dexamethasone was initiated but subsequently was withdrawn prior to patient enrollment. In addition, it should be noted that a planned phase I study (NCT01309412) of siltuximab in RRMM patients in Japan [103] had to be terminated because of unspecified safety issues. A trial (NCT01484275) to study siltuximab in high-risk smoldering MM patients remains active although patient recruitment has ceased. 

### 5.3. IL-15

IL-15, a member of the common γ-chain (γ_c_) receptor family of cytokines that includes IL-2 among others, induces antiapoptotic signaling in dendritic cells, cytotoxic T lymphocytes, and NK cells to enhance their survival. Signaling is mediated by trans presentation of membrane-bound IL15/IL-15Rα on activated monocytes and dendritic cells to surrounding cytotoxic T-cells and NK cells bearing the IL-15Rβ/γc surface receptor. Ligand binding activates target cell JAK/STAT, PI3K/Akt, and Ras/MAPK signaling pathways leading to enhanced cellular growth, inhibition of apoptosis, and augmented immune cell activation and migration [104]. The IL-15 superagonist ALT-803, a fusion protein engineered using a mutated IL-15 (N72D) linked to the IL-15Rα sushi domain and IgG1Fc, intensifies activation of cytotoxic T-cells and NK cells. Preclinical studies demonstrated that ALT-803 compared to IL-15 alone increased serum half-life and prolonged survival in myeloma-bearing mice [105,106]. ALT-803 is included in three ongoing myeloma-based trials: a phase I single agent dose-escalation study (NCT02099539) in RRMM and a phase II trial with elotuzumab and melphalan in high-risk MM patients following ASCT and infusion of enhanced NK cells (NCT03003728). In addition, two relapsed MM patients were included in a phase I/II trial (NCT01885897) to test the efficacy of ALT-803 in various hematologic malignancies (*n* = 33) following allogeneic SCT. A recent report [107] on this third trial concluded that ALT-803 is well-tolerated and significantly increased NK and CD8^+^ cell numbers although specific data relating to the MM patients was lacking. 

### 5.4. BCMA/BAFF/APRIL Axis

The cytokines BAFF (B-cell activating factor), sometimes referred to as B-lymphocyte stimulator (BLys or CD257), and its closely related homolog APRIL (a proliferation-inducing ligand), both members of the TNF superfamily, have received much attention in recent years centered around their crucial roles in the pathology of autoimmune diseases, such as lupus erythematosus and rheumatoid arthritis [108]. The production of both BAFF and APRIL by osteoclasts, monocytes, and neutrophils in the BMM also is considered a contributing factor to the proliferation and viability of myeloma cells [109,110,111,112]. Serum levels of BAFF, in particular, have been positively correlated with myeloma disease progression and prognosis [113,114,115]. BAFF and APRIL both serve as ligands for two transmembrane receptors on myeloma cells—TACI (transmembrane activator and calcium modulator and cyclophilin ligand interactor) and B-cell maturation antigen (BCMA). In addition, BAFF also binds to a third myeloma cell receptor, BAFF-R, while APRIL interacts with the sulphated side chains of heparan sulphate proteoglycan (HSPG) also on the myeloma cell surface [116]. Binding of BAFF and APRIL to these sites activates the NFκB, PI3K, and MAPK pathways to promote survival, dexamethasone resistance, and adhesion of myeloma cells to the BMSC [117,118,119].

Atacicept, an inhibitor of both BAFF and APRIL and the APRIL blocker tabalumab, each have been studied in several conditions, including MM, but have failed to provide evidence of efficacy or safety in any [120,121]. Meanwhile, BION-1301, a humanized anti-APRIL antibody, recently has emerged as a new possibility for clinical development in MM therapy [122] (NCT03340883). However, most of the focus on the BAFF/APRIL/BCMA axis in MM has been on BCMA as a major target of interest as evidenced by work on three immunotherapy fronts [123]: as a monoclonal ADC, as a component of the bispecific T-cell engager (BiTE) strategy (see Section 7), and in conjunction with chimeric antigenic receptor-T cell (CAR-T) therapy [124,125].

GSK2857916 (J6M0-mcMMAF) is a humanized afucosylated anti-BCMA antibody conjugated to monomethyl auristatin F, a microtubule inhibitor, via a non-cleavable protease-resistant maleimidocaproyl linker [126]. The antibody binds to the myeloma cell’s BCMA receptor to block BAFF and APRIL signaling while the auristatin component is released intracellularly via a lysosome-dependent mechanism causing cell cycle arrest at the G2/M checkpoint [127,128]. Preclinical studies demonstrated that GSK2857916 works to kill myeloma cells by virtue of its ability to cause ADCC, ADCP, and apoptosis, making this the first therapeutic ADC to work by three distinctly different mechanisms [126]. Currently, GSK2857916 is in the early stage of clinical development as the subject of a phase I study to determine the drug’s pharmacokinetic parameters, pharmacodynamic characteristics, therapeutic potential, and safety in RRMM patients (NCT02064387). As a single agent GSK2857916 exhibited a 60% ORR with a median PFS of 7.9 months in 35 heavily pretreated RRMM patients [129]. Corneal problems, thrombocytopenia, and anemia were cited as the most commonly observed adverse events. Based on these data, GSK2857916 recently was granted breakthrough therapy status for RRMM by the FDA, as well as PRIME designation from the European Medicines Agency.

Two additional anti-BCMA ADCs, MEDI2228 and AMG 224, are in ongoing phase I trials for RRMM—NCT03489525 and NCT02561962, respectively. In the former, BCMA is conjugated via a protease-cleavable pyrrolobenzodiazepine warhead linker, while AMG 224 is comprised of an antitumor maytansine derivative connected to antibody lysine residues via the non-cleavable 4-(*N*-maleimidomethyl) cyclohexane-1-carboxylate linker. 

### 5.5. CXCR4

The G-protein coupled chemokine receptor CXCR4 upon binding its ligand CXCR12, expressed by stromal cells, activates the PI3K/MAPK signal transduction pathway to promote cell growth, survival, proliferation, and migration. The CXCR4/CXCR12 axis has been shown to be a critical step in progression of several tumor types, including MM, in which increased expression of CXCR4 is linked to advanced metastatic disease with poor prognosis [130]. Animal studies have shown that the anti-CXCR4 antibody ulocuplumab (BMS-936564; MDX1338) blocked myeloma cell dissemination [131], providing a basis for a phase Ib study (NCT01359657) of the agent in combination with low-dose dexamethasone and either lenalidomide or bortezomib in RRMM patients. The ORR for the combined groups was 50% (22/44) with the lenalidomide arm exhibiting a somewhat better response rate than those receiving bortezomib (55% vs. 40%) [132].

### 5.6. CD137

CD137 (4-1BB), a co-stimulatory member of the TNF receptor superfamily, is expressed by several immune cells as a result of activation. These include T cells, dendritic cells, and NK cells in which it is known to potentiate ADCC and antigen-specific immune responses in T-cell directed therapy [133]. Binding of CD137 ligand, also a TNF family member, has been demonstrated to inhibit proliferation and induce apoptosis in MM cell lines [134], which has led to the search for CD137 agonists for potential use in MM. One such mAb is urelumab (BMS-663513) [135], a fully human IgG4 CD137 agonist whose safety as monotherapy has been demonstrated [136] and currently is the subject of several combination studies for both solid and hematologic tumors, including a phase I trial with elotuzumab in MM (NCT02252263). Utomilumab, another CD137 agonist, is the subject of patient studies in a number of cancers, although no trials appear to be in the offing for MM.

### 5.7. Antibodies Targeting Bone Loss in MM

MM commonly is accompanied by osteolytic bone disease, being present in an estimated 70–80% of patients upon initial diagnosis. Bone resorption and the resulting hypercalcemia are due to an imbalance between the activities of osteoclasts and osteoblasts whereby the former is upregulated while the latter is inhibited [137,138]. Competition between these two bone-remodeling processes is the result of a complex interplay chiefly involving four pathways, each of which has served as a major therapeutic target in efforts to reverse myeloma bone disease: the osteoclast stimulants RANK/RANKL and activin-A and the osteoblast antagonists DKK-1 and sclerostin [139].

#### 5.7.1. RANK/RANKL

Receptor-activator of nuclear κB ligand (RANKL), which is expressed by osteoblasts and bone marrow stem cells, is elevated in MM patients [140]. Interaction of RANKL with its receptor (RANK) activates osteoclasts via the NFκB pathway, resulting in bone resorption. Osteoprotegerin (OPG), a soluble member of the TNF receptor superfamily produced by osteoblasts, acts as a decoy receptor for RANKL to block osteoclast-mediated bone resorption [141]. Denosumab (AMG 162) is a humanized mAb that binds to RANKL to mimic the actions of OPG [142]. Originally approved in 2010 by the FDA for treatment and prevention of osteoporosis in postmenopausal women, as well as in prevention of skeletal-related events associated with bone metastases from solid tumors, denosumab’s list of indications was expanded in January 2018 to include MM. Approval was based on the results of a large-scale (1718 patients) phase III study (NCT01345019) that compared denosumab with the intravenously administered bisphosphonate zaledronic acid [143], heretofore, along with pamidronate, considered the standard treatment for skeletal-related events in MM [144]. Moreover, denosumab’s excretion by extra-renal pathways is seen as a distinct advantage over bisphosphonates whose elimination via the kidneys can be a major factor exacerbating renal complications of MM [145]. 

#### 5.7.2. Activin

Activin is a member of the TGFβ superfamily that was first discovered as a contributor to menstrual cycle regulation by stimulating FSH synthesis and secretion from the pituitary gland [146]. In addition to its other roles in normal physiology, activin, especially activin A, functions as an inhibitor of osteoblast-induced bone mineralization, as well as an enhancer of osteoclast activity [147]. Its role as a promoter of bone disease has made activin a prime target for drug development in osteoporosis and cancers, including MM [148]. Sotatercept (ACE-011), a fusion protein of the extracellular domain of activin receptor 2A and the Fc portion of IgG1, has been shown to prevent osteoporosis- and cancer-related bone loss in preclinical models [149]. In a phase IIA trial (NCT00747123), sotatercept, the humanized version of its mouse ortholog (RAP-011), has been shown to inhibit bone loss and increase hemoglobin levels while being well-tolerated in MM patients [150]. Preliminary results from a phase I trial (NCT01562405) of sotatercept with lenalidomide and dexamethasone provided similar outcomes [151].

#### 5.7.3. DKK-1 

The Wnt signaling pathway, which plays a major role in both embryogenesis and adult animal development, is a key regulator of osteoblastogenesis [152]. Dickkoff-1 (DKK-1), a soluble inhibitor of the canonical Wnt signaling pathway, is secreted by myeloma cells and acts as an inhibitor of osteoblast development [153,154]. Blood levels of DKK-1 have been shown to be elevated in myeloma patients [155] and are correlated with the extent of bone disease [156]. Furthermore, different anti-myeloma regimens are known to reduce DKK-1 serum levels in treatment responders [157], providing a basis for development of DKK-1-targeted antibodies for myeloma bone disease [158]. Based on preclinical studies [159,160] showing that BHQ880, a humanized IgG1 anti-DKK-1 mAb, inhibited myeloma cell-induced osteolytic lesion formation, as well as myeloma cell growth, led to a phase IB multi-center study of BHQ880 in combination with zoledronic acid and anti-myeloma therapy in 28 RRMM patients (NCT00741377). Although BHQ880 appeared to be well-tolerated, study design precluded direct interpretation of any ameliorating effects on bone lesions attributable to BHQ880. A trial of BHQ880 in patients with smoldering MM was recently completed but no results have been published as yet (NCT01302886). DKN-01 (LY2812176), a second humanized mAb DKK-1 inhibitor, has been studied in combination with lenalidomide-dexamethasone in RRMM but no detailed results have emerged from this trial (NCT01711671).

#### 5.7.4. Sclerostin

Another potential target for blocking bone loss in MM is sclerostin, which inhibits canonical Wnt signaling by binding to low-density lipoprotein receptor-related protein 5 (LRP5), a cell-surface receptor on osteoblasts, resulting in suppressed bone formation and concomitant predominance of osteoclastic activity to fuel bone loss. Increased sclerostin blood levels have been reported in MM patients compared to patients with MGUS and smoldering MM [161,162,163,164]. A recent study of myeloma cell lines and patient samples concluded that osteocytes, rather than malignant plasma cells, appear to be the source of sclerostin [165]. A number of clinical trials have studied the potential role of romosozumab, a humanized anti-sclerostin mAb, in osteoporosis but no studies have been initiated in MM patients and are not likely to in view of a recent report of serious cardiovascular events found in a significant proportion of patients receiving romosozumab in a phase III comparison with alendronate in osteoporosis. BPS804, another anti-sclerostin fully humanized mAb that has received orphan drug status for the treatment of osteogenesis imperfecta, has demonstrated the ability to reduce myeloma-induced bone loss in preclinical studies [165] although human trials of BPS804 in MM patients have yet to be initiated. 

## 6. Immune Checkpoint Inhibitors

### 6.1. PD-1 and PD-L1

Immune checkpoint blockade has become one of the most important strategies for overcoming barriers that prevent the body’s immune system from attacking tumor cells [166]. This has borne fruit most prominently as a result of work focused on the complex roles played by the PD-1/PD-L1, CTLA-4, and related immune checkpoint signaling pathways [167]. The Programmed Death-1 (PD-1) receptor and its cognate ligand (PD-L1) constitute a signaling axis that represents a well-established mechanism by which malignant cells block T cell-mediated immune responses to avoid recognition and elimination by the host immune system [168]. Tumor cells expressing PD-L1 on their surface bind to PD-1 on T lymphocytes to inhibit the latter’s proliferation and secretion of cytokines, as well as cause an increase in T regulatory cells. These combined effects not only produce immune tolerance to enable unrestrained tumor cell survival but also serve as the basis of an important strategic approach for anticancer immunotherapy [167]. Monoclonal antibodies that target the PD-1/PD-L1 immunological checkpoint have gained prominence in recent years with the FDA approval of several drugs, such as the PD-1 inhibitors pembrolizumab [169,170,171,172], nivolumab [173,174,175,176], and cemiplimab [177] and the PD-L1 blockers atezolizumab (https://www.fda.gov/drugs/informationondrugs/approveddrugs/ucm525780.htm (accessed on 25 September 2018)) and durvalumab [178] for Hodgkin’s lymphoma and a number of solid tumor types, including melanoma, bladder cancer, and squamous cell carcinomas of the head and neck. In a related and unprecedented development, the FDA recently granted accelerated approval to pembrolizumab for the treatment of solid tumors characterized by a microsatellite instability (mismatched repair deficiency), the first time a drug has been approved to treat a cancer based on a specific biomarker rather than on the organ of origin, a so-called tissue agnostic approach to cancer diagnosis and treatment (https://www.fda.gov/drugs/informationondrugs/approveddrugs/ucm560040.htm (accessed on 2 October 2018)). In March 2017, avelumab, a third PD-L1 blocker was approved as breakthrough therapy for metastatic Merkel cell carcinoma [179]. 

Blocking of immunological escape also has achieved a potentially important niche in the treatment of MM, in which initially promising results were obtained with some of the aforementioned mAbs [166,180,181]. PD-1 blocker pembrolizumab entered phase III trials in combination with dexamethasone and the immunomodulators lenalidomide (NCT02579863—KEYNOTE-185; patients newly diagnosed with MM) [182,183,184] or pomalidomide (NCT02576977—KEYNOTE-183; RRMM patients) [183,185,186,187]. However, in July 2017, the FDA (https://www.fda.gov/Drugs/DrugSafety/ucm574305.htm (accessed on 2 October 2018)) placed clinical holds on both studies due to the higher risk of death in the cohorts receiving the PD-1 blocker [183]. Moreover, in response to the safety concerns raised in these two trials, a third trial (NCT02036502; Keynote-023) [188,189,190], a phase I dose-finding study of pembrolizumab in combination with standard anti-myeloma agents, subsequently was placed on partial clinical hold (http://www.ascopost.com/News/57813 (accessed on 8 October 2018)). The FDA also placed partial or full holds on a number of other studies combining checkpoint inhibitors with immunomodulators, including the following: NCT02431208 (atezolizumab and/or daratumumab ± lenalidomide or pomalidomide (https://www.managedcaremag.com/news/20170919/fda-puts-two-more-cancer-drug-trials-partial-hold (accessed on 8 October 2018))) [191]; NCT02726581 (a phase III trial of nivolumab, pomalidomide, and dexamethasone); and NCT02685826 (durvalumab + lenalidomide ± dexamethasone (https://www.ashclinicalnews.org/news/fda-places-holds-clinical-trials-anti-pd-1-agent/ (accessed on 8 October 2018))). A myeloma-based phase I trial (NCT03357952) of the anti-PD-1 mAb JNJ-63723283 [192] in combination with daratumumab was recently discontinued for safety reasons. One myeloma trial of cemiplimab with isatuximab (NCT03194867) is ongoing while no trials of avelumab in MM patients have been initiated. Table 4 lists several of the currently active myeloma-based studies that include checkpoint inhibitors. 

### 6.2. CTLA-4

Costimulatory signals generated by engagement of CD28 on the surface of T_H_ cells with its ligand (B7) on antigen-presenting cells are blocked through a negative feedback mechanism whereby cytotoxic T-lymphocyte–associated protein 4 (CTLA-4) competes with the CD28-B7 binding to inhibit T cell activation. Monoclonal antibodies that bind to CTLA-4 thus have a stimulatory effect and should be capable of producing an anti-tumor immune response. Most notably, this strategic approach has been successfully applied to the immunotherapy of advanced melanoma by the anti-CTLA-4 mAb ipilimumab, approved by the FDA in 2011. A second mAb in this class, tremelimumab, has been studied primarily in mesothelioma although results here have been disappointing (http://www.pharmatimes.com/news/az_tremelimumab_fails_in_mesothelioma_trial_1020336 (accessed on 25 September 2018)). Two trials of CTLA-4 inhibitors in the context of MM are listed in the clinicaltrials.gov database (see Table 4). One small phase I study (NCT02716805) that included both durvalumab and tremelimumab has been placed on partial hold by the FDA due to unspecified additional data. Recruitment of both MM and lymphoma patients is currently in progress for a trial (NCT02681302) employing two checkpoint inhibitors—ipilimumab and nivolumab.

### 6.3. KIR

Killer-cell immunoglobulin-like receptors (KIR) are members of a family of transmembrane glycoproteins found on the surface of NK cells, as well as on some T cells. Both inhibitory and activating KIRs are known, the former acting to suppress NK cytotoxic actions when bound to MHC class I ligands on target cells, such as infected or transformed cells, enabling evasion of immune surveillance. IPH2101 (1-7F9), a mAb directed against three of the most commonly expressed inhibitory KIRs—KIR2DL-1, -2, and -3 [193], has been the subject of five myeloma-based trials (see Table 4). In one phase I trial (NCT01217203), IPH2101 was combined with lenalidomide in a steroid-free regimen that was well-tolerated although only 5 of the 16 RRMM patients showed clinical benefit [194]. In addition, IPH2101 as monotherapy unexpectedly produced a drop in NK cell responsiveness in a phase II study (NCT01248455) in patients with smoldering MM, leading to trial termination [195]. The second-generation anti-KIR agent lirilumab (BMS-986015; IPH2102) currently is included in one arm of a myeloma-based phase I trial (NCT02252263) with elotuzumab, although no results have been made public to date.

### 6.4. CD47

CD47 is a ubiquitously-expressed integrin-associated cell surface protein whose interaction with signal regulatory protein-α (SIRPα), found on the surface of phagocytic cells, most notably macrophages, serves as a “don’t eat me” signal, protecting these cells from engulfment [196]. Upregulation of CD47 by different cancer cell types, including 73% of patient-derived myeloma cells [197], is an important mechanism that enables tumor cells to avoid detection and destruction by the immune system. Thus, CD47-blocking antibodies have come to the fore in recent years as a new class of checkpoint inhibitors of potential use in treatment of hematological malignancies [198,199]. These include TTI-621 and TTI-622 under development by Trillium Therapeutics. TTI-621, a recombinant fusion protein comprised of the SIRPα binding domain linked to the Fc segment of human IgG1, is currently included in phase I trials for both solid tumors (NCT02890368) and blood cancers (NCT02663518). Recruitment for a study of TTI-622, similar to TTI-621 but linked to IgG4, in RRMM and lymphoma patients (NCT03530683) recently has been initiated. Reports that activity of both TTI-621 and TTI-622 against myeloma cells in animal models can be enhanced by combination with a proteasome inhibitor have informed their inclusion in arms of these trials [200]. In addition, SRF231, a fully human anti-CD47 mAb whose construction details have yet to be disclosed, was recently granted orphan drug status by the FDA and is the subject of a phase I study (NCT03512340) in patients with both solid and hematologic cancers.

## 7. BiTE^®^ Antibodies

Bispecific T-cell engager (BiTE^®^) antibodies represent an innovative immunotherapeutic approach to cancer treatment. The method developed by Micromet AG in collaboration with Amgen already has yielded success in the form of the CD3–CD19 construct blinatumomab, which in 2014 received FDA approval for use in Philadelphia chromosome-negative B-cell precursor acute lymphocytic leukemia [204]. MM-based clinical work with blinatumomab has been limited to one ongoing pilot study (NCT03173430) in which the agent is combined with salvage ASCT in RRMM. The BiTE^®^ strategy differs from regular T-cell mediated cytotoxicity by circumventing the need for antibody-presenting cells, costimulatory molecules, or the MHC/antigen complex in order to kill tumor cells [205]. In its application to MM, the method involves the construction of a recombinant antibody to two different epitopes – the CD3ε molecules on tumor-specific T cells and a tumor-specific receptor on the myeloma cell. Cross-linkage of the T-cells to the tumor cells via the created synapse causes the T-cells to release two cytolytic-initiating proteins: perforin, which oligomerizes to create transmembrane pores in the target cell, and the procaspase activator granzyme B, which traverses these pores to initiate apoptosis in the myeloma cells [205]. Most myeloma-based work in this field has revolved around antibodies designed to link T-cell CD3 with BCMA. For example, the BiTE^®^-based antibody BI-836909 (AMG 420), which exhibited good efficacy and potency in preclinical studies by causing depletion of BCMA-positive myeloma cells [206], is now the subject of a phase I dose-escalation study as monotherapy in RRMM patients (NCT02514239) (https://ash.confex.com/ash/2018/webprogram/Paper109769.html (accessed on 17 November 2018)). JNJ-64007957 (NCT03145181), CC-93269/EM901 (NCT03486067), AMG701 (NCT03287908), and PF-06863135 (NCT03269136) represent four additional CD3-BCMA constructs now in phase I clinical development for RRMM. Favorable anti-myeloma activity has been reported for EM801, another CD3epsilon-BCMA BiTE-based molecule with immunotherapeutic potential. In this case, T cell-myeloma cell linkage is followed by CD4^+^/CD8^+^ activation to cause secretion of interferon-γ, in addition to perforin and granzyme B [207]. Clinical trials of EM801 in MM patients have not yet begun. In addition to BCMA, two other myeloma surface antigens have shown preclinical promise as CD3 epitope binding partners, serving as the basis of recently initiated RRMM trials—the G-protein coupled receptor family C group 5 member D (GPRC5D) [208], incorporated into JNJ-64407564 (NCT03399799), and the Fc receptor-like protein 5 (FcRH5) (71022)[209], integrated into BFCR4350A (NCT03275103). A BiTE linking CD3 and CD138 (syndecan-1) has shown promise for MM in preclinical studies [210,211].

## 8. Additional mAbs

CD33, which is found primarily on myeloid cells, serves as the therapeutic target for gemtuzumab ozogamicin, an ADC approved for the treatment of acute myeloid leukemia. This transmembrane receptor also has attracted interest as a potential target for MM since it is expressed on 20–35% of myeloma cells, leading to development of lintuzumab-actinium 225, an α-emitting radioimmunoconjugate now in a phase I clinical trial (NCT02998047) for RRMM [212].

CD74 represents the invariant chain of an MHC Class II transmembrane glycoprotein that associates with the α and β chains of HLA-DR and, in addition to other roles, functions as a survival receptor by activating the NFκB signaling pathway [213]. Expression of CD74 is particularly elevated in B cell malignancies, such as non-Hodgkin’s lymphomas and MM [214]. The humanized anti-CD74 mAb milatuzumab (IMMU-115; hLL1) was the subject of a phase I monotherapy study in 25 RRMM patients that yielded no objective responses [215] (NCT00421525). Based on preclinical studies showing the efficacy of combining milatuzumab with an anthracycline [216], the conjugate of milatuzumab with doxorubicin (IMMU-110; hLL1-DOX) was prepared and is now the subject of a phase I/II study in myeloma patients (NCT01101594). Another anti-CD74 drug is STRO-001 in which an aglycosylated human IgG1 antibody (SP7219) is conjugated to p-azido-methyl-phenylalanine (pAMF) via a non-cleavable dibenzocyclooctyne (DBCO)-maytansinoid linker-warhead. A clinical trial (NCT03424603) of this ADC in patients with advanced B-cell cancers, including MM, was initiated in early 2018. In October 2018, the FDA granted STRO-001 orphan drug status for MM.

Neural cell adhesion molecule (NCAM or CD56) is a member of the immunoglobulin superfamily. Although not expressed in normal plasma cells, this membrane glycoprotein is strongly upregulated in malignant cells from most MM patients [217,218]. The ADC lorvotuzumab mertansine (HuN901-DM1; IMGN901; BB-10901) represents an anti-NCAM humanized IgG1 mAb linked to a maytansine derivative with potent tubulin assembly inhibitory activity. A phase I study of the conjugate combined with lenalidomide and dexamethasone in 44 CD56^+^ RRMM patients has been concluded [219]. The ORR of 59% was considered insufficiently robust and, combined with the emergence of dose-related peripheral neuropathy as a significant adverse event, reportedly has led to discontinuation of further studies of this conjugate [220].

Syndecan-1 (CD138), a single-pass transmembrane member of the heparan sulfate proteoglycan family, mediates cell adhesion to collagen and fibronectin in the ECM. In addition, it serves to bind ligands of the epidermal growth factor family, fibroblast growth factor, and hepatocyte growth factor [221,222]. Although generally found on the surface of epithelial cells, in the hematopoietic system this receptor is confined exclusively to plasma cells, where its overexpression in myeloma cells relative to normal plasma cells has made CD138 a target for drug discovery [210,223,224]. Indatuximab ravtansine (BT062) is an anti-CD138 mAb conjugated to an anti-tubular maytansoid (DM-4) derivative through a disulfide linker [225]. Preclinical work with this ADC in both myeloma cell lines and in vivo mouse xenograft models showed enhanced activity with lenalidomide and dexamethasone [226], leading to initiation of a phase I/II trial (NCT01638936) of the three-drug combination in RRMM, resulting in an ORR of 78% in 64 RRMM patients who had received a median of three prior therapies [227]. Replacement of lenalidomide by pomalidomide produced no significant change in ORR (79%) [228]. Fatigue, hypokalemia, nausea, and diarrhea were the most common adverse effects reported in these studies.

## 9. Summary 

The introduction of mAbs into the fight against MM represents one of the most significant therapeutic advances made against the disease in the past decade, joining proteasome inhibitors, immunomodulators, HDAC blockers, alkylating agents, and corticosteroids as effective regimens for the disease. Identification of CD38 and SLAMF7 as suitable therapeutic targets because of their high expression on the surfaces of malignant plasma cells led to the development of daratumumab and elotuzumab, respectively, both of which received FDA approval as anti-myeloma agents three years ago. Network meta-analytical studies of RCTs that include daratumumab and an immunomodulator thus far have yielded impressive outcomes in the RRMM setting. Moreover, this mAb also has shown some potential in the context of frontline therapy of MM, as well as in high-risk smoldering myeloma. Although clinical results with elotuzumab in RRMM have been to some extent less striking, this mAb has been shown to be efficacious when included with dexamethasone and either lenalidomide or bortezomib. The results of several ongoing trials of these two mAbs, as well as of studies incorporating the newer anti-CD38 antibodies isatuximab and MOR-202 may offer the possibility of future improvements in the disease’s outcomes, whether as initial therapies or in the setting of relapsed-refractory disease. 

Much interest has been drawn in recent years to BCMA as an attractive target for myeloma-directed immunotherapy [123]. One agent in particular, the BCMA-ADC GSK2857916, has yielded impressive results in early trials, demonstrating efficacy, as well as an acceptable safety profile in patients for whom standard myeloma-based therapies are no longer effective [129]. In addition, innovative bispecific T-cell engager (BiTE^®^) technology employing a number of BCMA constructs have recently entered clinical trials for RRMM. Although not a focus of this review, note should be made of the potential of BCMA-based CAR-T therapy to make sizeable future inroads into cancer immunotherapy, including MM [229]. Efforts in this arena continue unabated in spite of substantial challenges [123]. 

Although checkpoint inhibitors have been successfully introduced for the treatment of several cancers [167] and they initially showed great promise in RRMM, the future viability of this group of agents in the disease appears clouded at present. This uncertainty is due to data from two phase III trials that revealed a higher number of deaths in myeloma patient cohorts in which checkpoint inhibitors were combined with immunomodulators, causing the FDA to order clinical holds on these and related studies [183].

Meanwhile, efforts to discover mAbs capable of treating myeloma-associated bone loss have shown progress. A major trial comparing denosumab with zoledronic acid confirmed that both agents are comparable in terms of preventing skeletal-related events in MM, leading to the recent approval by the FDA of denosumab for this purpose. The reduced renal toxicity associated with denosumab, due to its elimination by extra-renal pathways, appears to offer a distinct advantage over the more established zoledronic acid in this setting. 

The past two decades have witnessed a surge of positive developments in the search for new drugs to treat MM. These efforts have resulted not only in a deeper understanding of the mechanisms by which this unrelenting disease progresses but also in the identification of suitable molecular targets relevant to those mechanisms. The era of immunotherapy in the treatment of MM was inaugurated in 2015 with the regulatory approval of mAbs targeting CD38 and SLAMF7 and the future looks bright for other immunological agents to follow in their wake as MM continues its transition from a disease heretofore considered incurable to one more resembling a manageable chronic condition. 

## Figures and Tables

**Table 1 ijms-19-03924-t001:** Selected Active Trials of Daratumumab in Multiple Myeloma (MM).

Trial ID [References]	Treatment	Phase	Enrollment	Trial Title
NCT03695744	Dara + Bort + Dex	II	63	AMN006—Phase 2 Study of Daratumumab in Combination with Bortezomib and Dexamethasone in Newly Diagnosed Transplant Ineligible Multiple Myeloma Patients
NCT03158688	Dara + Carf + Dex	III	466	A Randomized, Open-label, Phase 3 Study Comparing Carfilzomib, Dexamethasone, and Daratumumab to Carfilzomib and Dexamethasone for the Treatment of Patients with Relapsed or Refractory Multiple Myeloma (CANDOR)
NCT03180736 [26]	Dara + Pom + Dex	III	302	A Phase 3 Study Comparing Pomalidomide and Dexamethasone with or without Daratumumab in Subjects with Relapsed or Refractory Multiple Myeloma Who Have Received at Least One Prior Line of Therapy with Both Lenalidomide and a Proteasome Inhibitor.
NCT02541383	Dara + Bort + Thal + Dex	III	1085	Study of Daratumumab in Combination with Bortezomib (VELCADE), Thalidomide, and Dexamethasone (VTD) in the First Line Treatment of Transplant Eligible Subjects with Newly Diagnosed Multiple Myeloma
NCT03277105 [27]	Dara	III	480	A Phase 3 Randomized, Multicenter Study of Subcutaneous vs. Intravenous Administration of Daratumumab in Subjects with Relapsed or Refractory Multiple Myeloma
NCT02136134 [10,28]	Dara + Bort + Dex	III	499	Phase 3 Study Comparing Daratumumab, Bortezomib and Dexamethasone (DVd) vs. Bortezomib and Dexamethasone (Vd) in Subjects with Relapsed or Refractory Multiple Myeloma (CASTOR)
NCT02076009 [29]	Dara + Len + Dex	III	569	Phase 3 Study Comparing Daratumumab, Lenalidomide, and Dexamethasone (DRd) vs. Lenalidomide and Dexamethasone (Rd) in Subjects with Relapsed or Refractory Multiple Myeloma (POLLUX)
NCT03217812	Dara + Bort + Mel + Pred	III	210	A Phase 3, Multicenter, Randomized, Controlled, Open-label Study of VELCADE (Bortezomib) Melphalan-Prednisone (VMP) Compared to Daratumumab in Combination with VMP (D-VMP), in Subjects with Previously Untreated Multiple Myeloma Who Are Ineligible for High-Dose Therapy (Asia Pacific Region)
NCT02195479 [17]	Dara + Mel + Bort + Pred/Dex	III	706	A Phase 3, Randomized, Controlled, Open-label Study of VELCADE (Bortezomib) Melphalan-Prednisone (VMP) Compared to Daratumumab in Combination with VMP (D-VMP), in Subjects with Previously Untreated Multiple Myeloma Who Are Ineligible for High-Dose Therapy (ACYONE)
NCT03475628	Dara	II	57	A Prospective, Multicenter, Non-comparative, Open-label, Phase II Study to Evaluate the Effects of Daratumumab Monotherapy on Bone Parameters in Patients with Relapsed and/or Refractory Multiple Myeloma Who Have Received at Least 2 Prior Lines of Therapy, Including Lenalidomide and a Proteasome Inhibitor
NCT02626481 [30]	Dara + Dex	II	64	A Multicenter Open Label Phase II Study of Daratumumab in Combination with Dexamethasone in Multiple Myeloma Resistant or Refractory to Bortezomib and Lenalidomide and Pomalidomide—an IFM 2014-04 Study
NCT02316106 [18]	Dara	II	126	A Randomized Phase 2 Trial to Evaluate Three Daratumumab Dose Schedules in Smoldering Multiple Myeloma (CENTAURUS)
NCT03301220 [19]	Dara	III	360	A Phase 3 Randomized Multicenter Study of Subcutaneous Daratumumab Versus Active Monitoring in Subjects with High Risk Smoldering Multiple Myeloma (AQUILA)

Bort = bortezomib; Carf = carfilzomib; Dara = daratumumab; Dex = dexamethasone; Len = lenalidomide; Mel = melphalan; Pom = pomalidomide; Pred = prednisone; Thal = thalidomide.

**Table 2 ijms-19-03924-t002:** Active Trials of Isatuximab in MM.

Trial ID [References]	Treatment	Phase	Enrollment	Trial Title
NCT02960555 [25]	Isatux	II	61	Phase II Single Arm Trial of Isatuximab (SAR650984) in Patients with High Risk Smoldering Multiple Myeloma (ICARIA)
NCT02812706 [25]	Isatux	I/II	42	A Phase I/II Study of Isatuximab (Anti-CD38 mAb) Administered as a Single Agent in Japanese Patients with Relapsed and Refractory Multiple Myeloma (Islands)
NCT02514668 [25,31]	Isatux	I	64	An Open-label, Dose-escalation and Multi-center Study to Evaluate the Safety, Pharmacokinetics and Efficacy of SAR650984 (Isatuximab) in Patients with Relapsed/Refractory Multiple Myeloma
NCT03194867	Isatux + Cemip	I/II	105	Phase 1/2 Study to Evaluate Safety, Pharmacokinetics and Efficacy of Isatuximab in Combination with Cemiplimab in Patients with Relapsed/Refractory Multiple Myeloma
NCT03275285	Isatux + Carf + Dex	III	300	Randomized, Open Label, Multicenter Study Assessing the Clinical Benefit of Isatuximab Combined with Carfilzomib (Kyprolis®) And Dexamethasone versus Carfilzomib with Dexamethasone in Patients with Relapse and/or Refractory Multiple Myeloma Previously Treated with 1 to 3 Prior Lines (IKEMA)
NCT02990338 [25]	Isatux + Pom + Dex	III	300	A Phase 3 Randomized, Open-label, Multicenter Study Comparing Isatuximab (SAR650984) in Combination with Pomalidomide and Low-Dose Dexamethasone versus Pomalidomide and Low-Dose Dexamethasone in Patients with Refractory or Relapsed and Refractory Multiple Myeloma (ICARIA-MM)
NCT02513186	Isatux + Len + Bort + Dex + Cp	I	44	A Dose Escalation, Safety, Pharmacokinetic, Pharmacodynamic and Preliminary Efficacy Study of SAR650984 (Isatuximab) Administered Intravenously in Combination with Bortezomib—Based Regimens in Adult Patients with Newly Diagnosed Multiple Myeloma Non-Eligible for Transplantation CyBorDSAR)
NCT03617731	Isatux + Bort + Dex + Len	III	662	A Randomized Phase III Trial Assessing the Benefit of the Addition of Isatuximab to Lenalidomide/Bortezomib/Dexamethasone (RVd) Induction and Lenalidomide Maintenance in Patients with Newly Diagnosed Multiple Myeloma (GMMG HD7)
NCT01749969 [24]	Isatux + Len + Dex	I	60	A Phase 1b Study of SAR650984 (Anti-CD38 mAb) in Combination with Lenalidomide and Dexamethasone for the Treatment of Relapsed or Refractory Multiple Myeloma
NCT03104842	Isatux + Len + Dex + Carf	II	153	Clinical Phase II, Multicenter, Open-label Study Evaluating Induction, Consolidation and Maintenance with Isatuximab (SAR650984), Carfilzomib, Lenalidomide and Dexamethasone (I-KRd) in Primary Diagnosed High-risk Multiple Myeloma Patients
NCT02283775 [22,23]	Isatux + Pom + Dex	I	89	A Phase 1b Study of SAR650984 (Isatuximab) in Combination with Pomalidomide and Dexamethasone for the Treatment of Relapsed/Refractory Multiple Myeloma (PomdeSAR)
NCT03319667	Isatux + Bort + Len + Dex	III	440	A Phase 3 Randomized, Open-label, Multicenter Study Assessing the Clinical Benefit of Isatuximab (SAR650984) in Combination with Bortezomib (Velcade^®^), Lenalidomide (Revlimid^®^) and Dexamethasone versus Bortezomib, Lenalidomide and Dexamethasone in Patients with Newly Diagnosed Multiple Myeloma (NDMM) Not Eligible for Transplant (IMROZ)

Bort = bortezomib; Carf = carfilzomib; Cemip = cemiplimab; Cp = cyclophosphamide; Dex = dexamethasone; Isatux = isatuximab; Len = lenalidomide; Pom = pomalidomide.

**Table 3 ijms-19-03924-t003:** Selected Active Trials of Elotuzumab in MM.

Trial ID [References]	Treatment	Phase	Enrollment	Trial Title
NCT01891643/NCT01335399	Elo + Dex + Len	III	750	A Phase 3, Randomized, Open Label Trial of Lenalidomide/Dexamethasone with or without Elotuzumab in Subjects with Previously Untreated Multiple Myeloma
NCT02495922 [45]	Elo + Len + Bort + Dex	III	564	A Randomized Phase III Trial on the Effect of Elotuzumab in VRD Induction /Consolidation and Lenalidomide Maintenance in Patients with Newly Diagnosed Myeloma
NCT01478048 [46]	Elo + Bort + Dex	II	185	A Phase 2, Randomized Study of Bortezomib/Dexamethasone with or without Elotuzumab in Subjects with Relapsed/Refractory Multiple Myeloma
NCT03361306	Elo + Len + Carf + Dex	II	40	LCI-HEM-MYE-CRD-002: A Phase II Study of Carfilzomib-Revlimid-Dexamethasone-Elotuzumab in Relapsed/Refractory Multiple Myeloma
NCT01668719 [47,48]	Bort + Len + Dex ± Elo	I/II	122	A Randomized Phase I/II Study of Optimal Induction Therapy of Bortezomib, Dexamethasone and Lenalidomide with or without Elotuzumab (NSC-764479) for Newly Diagnosed High Risk Multiple Myeloma (HRMM)
NCT01239797 [36,49]	Len + Dex ± Elo	III	761	Phase 3, Randomized, Open Label Trial of Lenalidomide/Dexamethasone with or without Elotuzumab in Relapsed or Refractory Multiple Myeloma (MM)—(ELOQUENT-2)
NCT02654132 [37]	Elo + Pom + Dex	II	157	An Open Label, Randomized Phase 2 Trial of Pomalidomide/Dexamethasone with or Without Elotuzumab in Relapsed and Refractory Multiple Myeloma (ELOQUENT-3)
NCT03168100	Elo + Bort + Len + Dex	II	115	A Single-Arm, Open-label Study of Anti-SLAMF7 mAb Therapy After Autologous Stem Cell Transplant in Patients with Multiple Myeloma
NCT03393273	Elo	II	35	Induction and Consolidation with Elotuzumab before and after Peripheral Stem Cell Autologous Graft in Elderly Patients with Multiple Myeloma
NCT02718833 [50]	Elo + Pom + Bort + Dex	II	46	Phase II Study of Elotuzumab in Combination with Pomalidomide, Bortezomib, and Dexamethasone in Relapsed and Refractory Multiple Myeloma
NCT02159365 [51]	Elo + Len + Dex	II	81	A Phase 2 Single Arm Study of Safety of Elotuzumab Administered Over Approximately 60 Minutes in Combination with Lenalidomide and Dexamethasone for Newly Diagnosed or Relapsed/Refractory Multiple Myeloma Patients

Bort = bortezomib; Carf = carfilzomib; Elo = elotuzumab; Dex = dexamethasone; Len = lenalidomide; Pom = pomalidomide.

**Table 4 ijms-19-03924-t004:** Selected Trials of Checkpoint Inhibitors in MM ^a^.

Trial ID [References]	Treatment	Phase	Enrollment	Trial Title
**PD-1 Inhibitors**				
NCT02603887	Pembro	I	20	Pilot Single Arm, Single Center, Open Label Trial of Pembrolizumab in Patients with Intermediate and High Risk Smoldering Multiple Myeloma
NCT02906332	Pembro + Len + Dex	II	16	A Phase II Trial of the Anti -PD-1 Monoclonal Antibody Pembrolizumab (MK-3475) + Lenalidomide + Dexamethasone as Post Autologous Transplant Consolidation in Patients with High-risk Multiple Myeloma
NCT03221634	Pembro + Dara	II	57	A Phase 2 Study of Pembrolizumab in Combination with Daratumumab (Anti CD38) in Participants with Relapsed Refractory Multiple Myeloma (rrMM)
NCT03506360	Pembro + Ixazo + Dex	II	41	Phase 2 Trial of Pembrolizumab, Ixazomib, and Dexamethasone for Relapsed Multiple Myeloma
NCT02880228	Pembro + Len + Dex	II	41	Phase 2 Trial of Pembrolizumab, Lenalidomide, and Dexamethasone for Initial Therapy of Newly Diagnosed Multiple Myeloma Eligible for Stem Cell Transplantation
NCT02636010	Pembro	II	20	Phase II, Multicenter, Open Label, Clinical Trial of the Anti-PD1 Monoclonal Antibody Pembrolizumab (MK3475) as Consolidation Therapy in Multiple Myeloma Patients with Residual Disease After Treatment
NCT03267888	Pembro + Rad	I	24	Pilot Study of Pembrolizumab and Single-Fraction, Low-Dose, Radiation Therapy in Patients with Relapsed or Refractory Multiple Myeloma
NCT02331368	Pembro + Len + Mel + ASCT	II	32	Phase 2 Multi-center Study of Anti-Programmed-Death-1 [Anti-PD-1] During Lymphopenic State After High-Dose Chemotherapy and Autologous Hematopoietic Stem Cell Transplant [HDT/ASCT] for Multiple Myeloma
NCT03292263	Nivol + Mel + ASCT	I/II	30	Autologous Stem Cell Transplantation with Nivolumab in Patients with Multiple Myeloma
NCT03333746	Nivol + Len	II	18	Phase II Study of Lenalidomide in Combination with Nivolumab In Patients with Relapsed/Refractory Multiple Myeloma
NCT02726581 [201]	Nivol + Elo + Pom + Dex	III	348	An Open-Label, Randomized Phase 3 Trial of Combinations of Nivolumab, Pomalidomide and Dexamethasone in Relapsed and Refractory Multiple Myeloma
NCT02612779	Nivol + Elo + Pom + Dex	II	95	A Phase 2, Multiple Cohort Study of Elotuzumab in Combination with Pomalidomide and Low-Dose Dexamethasone (EPd), and in Combination with Nivolumab (EN), in Patients with Multiple Myeloma or Refractory to Prior Treatment with Lenalidomide
NCT03184194	Nivol + Dara + Cp	II	60	A Phase 2 Study of Nivolumab Combined with Daratumumab with or Without Low-dose Cyclophosphamide in Relapsed/Refractory Multiple Myeloma
NCT03605719	Nivol + Carf + Pom + Dex + Reo	I	62	PD1 Blockade and Oncolytic Virus in Relapsed Multiple Myeloma
NCT03634800	Nivol + Rad	II	30	Radiotherapy with Immunotherapy for Systemic Effect in Myeloma (RISE-M)
NCT03194867	Cemip + Isatux	I/II	105	Phase 1/2 Study to Evaluate Safety, Pharmacokinetics and Efficacy of Isatuximab in Combination with Cemiplimab in Patients with Relapsed/Refractory Multiple Myeloma
**PD-L1 Inhibitors**				
NCT02807454	Durva+ Dara + Pom + Dex	II	37	A Phase 2, Multicenter, Open-label, Study to Determine the Safety and Efficacy for the Combination of Durvalumab (DURVA) and Daratumumab (DARA) (D2) in Subjects with Relapsed and Refractory Multiple Myeloma (RRMM)
NCT02616640 [202]	Durva + Pom + Dex	I	114	A Phase IB Multicenter, Open-label Study To Determine The Recommended Dose And Regimen Of Durvalumab (MEDI4736) Either As Monotherapy or In Combination With Pomalidomide (POM) With Or Without Low-Dose Dexamethasone (DEX) In Subjects With Relapsed And Refractory Multiple Myeloma (RRMM)
NCT02784483 (suspended)	Atez	I	20	Pilot Study of Anti-Programmed Death Ligand-1 (Anti-PD-L1, Atezolizumab In Asymptomatic Myeloma
NCT03312530	Atez + Combi + Venet	I/II	72	A Phase Ib/II Study of Cobimetinib Administered as Single Agent and in Combination with Venetoclax, With or Without Atezolizumab, in Patients With Relapsed and Refractory Multiple Myeloma
**CTLA-4 Inhibitor**				
NCT02681302	Ipil + Nivol	I/II	42	Phase Ib-IIA Study of Combined Check Point Inhibition After Autologous Hematopoietic Stem Cell Transplantation in Patients at High Risk for Post-transplant Recurrence
**KIR Inhibitors**				
NCT01222286	IPH2101	II	30	Multicenter Phase II Study on the Anti-tumor Activity, Safety and Pharmacology of Two Dose Regimens of IPH2101, a Fully Human Monoclonal Anti-KIR Antibody, in Patients with Smoldering Multiple Myeloma (KIRMONO)
NCT00999830	IPH2101	II	27	Randomized Phase II Study Evaluating the Anti-tumor Activity, Safety and Pharmacology of Two Dose Regimens of IPH2101, a Human Monoclonal Anti-KIR Antibody, in Patients with Multiple Myeloma in Stable Partial Response After a First Line Therapy
NCT00552396 [203]	IPH2101	I	32	An Open-label, Dose-escalation Safety and Tolerability Trial Assessing Multiple Dose Administrations of Anti-KIR (1-7F9) Human Monoclonal Antibody in Subjects with Multiple Myeloma
NCT01217203 [194]	IPH2101	I	15	Multicenter Phase I Study on the Safety, Anti-tumor Activity and Pharmacology of IPH2101, a Human Monoclonal Anti-KIR, Combined with Lenalidomide in Patients with Multiple Myeloma Experiencing a First or Second Relapse
NCT02252263	Liri + Elo + Urel	I	44	A Phase I Open Label Dose Escalation and Randomized Cohort Expansion Study of the Safety and Tolerability of Elotuzumab (BMS-901608) Administered in Combination with Either Lirilumab (BMS-986015) or Urelumab (BMS-663513) in Subjects with Multiple Myeloma
**CD47 Inhibitors**				
NCT03530683	TTI-622 + (Bort or Carf) + Dex	I	156	A Phase 1a/1b Dose Escalation and Expansion Trial of TTI-622 in Patients with Advanced Relapsed or Refractory Lymphoma or Myeloma

^a^ Unless otherwise stated, includes only studies devoted specifically to MM patients. ASCT = autologous stem cell transplantation; Atez = atezolizumab; Bort = bortezomib; Carf = carfilzomib; Cemip = Cemiplimab; Combi = cobimetinib; Cp = cyclophosphamide; Dara = daratumumab; Dex = dexamethasone; Durva = durvalumab; Elo = elotuzumab; Ipil = ipilimumab; Ixazo = ixazomib; Len = lenalidomide; Liri = lirilumab; Mel = melphalan; Nivol = nivolumab; Pembro = pembrolizumab; Pom = pomalidomide; Rad = radiation; Reo = reovirus; Urel = urelumab; Venet = venetoclax.

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
