# Peer review of "Monoclonal Antibodies for the Treatment of Multiple Myeloma: An Update"

_ijms, 2018, doi:10.3390/ijms19123924_

Reviewer 1 Report

In this paper HN Abramson reported an update on therapies based on the use of monoclonal antibodies for the treatment of multiple myeloma. The manuscript accurately describes the current therapies used, especially introducing the most recent works and the additional monoclonal antibodies now in the developmental pipeline. The review is well written, the study is very interesting, exhaustive and comprehensive. It could represent a scientific report useful for future oncology studies in multiple myeloma, especially in the clinical setting. According to me, the manuscript can be considered acceptable in the present form.

Author Response

The author thanks the reviewer for taking the time to review the manuscript. The comments are appreciated.

Reviewer 2 Report

This is a comprehensive and meticulous review that is well written and timely. As stated by the author, the discussion is divided primarily based on the molecular targets of each mAb. The even-handed approach based on target and FDA-approval however, fails to educate the reader on the current and future clinical uses of these compounds. For example, almost equal weight is given to promising antibody targets and targets whose products have failed. Who is the audience for this manuscript? If the answer is pharmacists, then there is little fault with the manuscript in its current form. However, this reviewer is a clinician-scientist, and I believe the manuscript could be improved with input from a clinical investigator and organization that reflects the current roles of these agents and the direction that this use is likely to evolve with time.

Line 41: “To this was added autologous stem cell transplantation…” Certainly it’s a mainstay of therapy, but here and later it is referred to in a way that emphasizes its novelty, but ASCT is basically a way to give higher doses of good old melphalan.

Line 52: “…have begun to bear fruit with the FDA approval in 2015..” This sentence ignores the spectacular clinical success of daratumumab. Many cancer drugs become FDA approved with minimal documented benefit, so it’s worth noting. The manuscript generally favors the status of an mAb vis a vis the FDA rather than the degree of benefit obtained.

Line 101: “..isatuximab may be better tolerated..” Is this true? Dara is very well tolerated. Mechanisms for how isa could be more potent than dara are presented, but I don’t understand how there would be less toxicity, and I did not find note of this in the Martin et al reference.

It is not mentioned that the dramatic success of dara in RRMM has lead to the question of whether it should be included as up front therapy, or in patients with smoldering MM. A tantalizing possibility that even deeper responses could be obtained up front.

Line 133: “..a total of 45 myeloma-based elotuzumab studies, either completed or in progress.” Clearly this and other drugs are coming to the clinic, but with little guidance on their proper role. Dimopoulos et al, New Engl J Med, 2018 just published one of these studies, which added no useful information. Missing is discussion on whether a given agent provides survival benefit over other treatments.

Line 276: "IL-6." Many targets might be combined into a section titled something like, “Failed antibody targets.” E.g. PD-1, PD-L1, DKK-1, CD137, IGF1R, and Sclerostin. How many of the trials on the 2.5 page long Table 4 are failed or aborted?

Line 580-581: “…has shown promise for MM in preclinical studies.” There is tremendous excitement about these agents. See https://ash.confex.com/ash/2018/webprogram/Paper109769.html

Line 622-623: “…the future looks bright for other immunological agents to follow in their wake as MM is transformed from an incurable disease to one more resembling a manageable chronic illness.” This transformation has already happened. For most patients diagnosed in 2018, MM is a chronic disease. The primary issue now is the difficult clinical trial work needed to deliver “the cure” that many in the field believe is on the horizon. We have an embarrassment of riches in the myeloma field regarding antibody therapies, and something of a Wild West approach at present in the clinic. There is little consensus on how to use these agents currently, and some direction would be appreciated.

Minor points

Line 31-32: “non-functional paraproteins” Generally this characterization is true of course, but it ignores a) evidence that monoclonal immunoglobulins may target common infectious agents, i.e. Bosseboeuf et al, JCI Insight 2017, and b) the toxic nature of some light chains, e.g. with the development of amyloidosis.

Author Response

The author thanks the reviewer for taking the time to read and offer several insightful comments on the manuscript.

Line 41: “To this was added autologous stem cell transplantation…” Certainly it’s a mainstay of therapy, but here and later it is referred to in a way that emphasizes its novelty, but ASCT is basically a way to give higher doses of good old melphalan.

Response: Each reference to ASCT was considered in light of the reviewer’s comment and was removed in cases where it is unnecessarily mentioned.

Line 52: “…have begun to bear fruit with the FDA approval in 2015..” This sentence ignores the spectacular clinical success of daratumumab. Many cancer drugs become FDA approved with minimal documented benefit, so it’s worth noting. The manuscript generally favors the status of an mAb vis a vis the FDA rather than the degree of benefit obtained.

Response: The daratumumab discussion has been expanded to include five recently published network meta-analytic studies highlighting the outcome successes realized by inclusion of this mAb in RRMM patient trials (see lines 79-91).

Line 101: “..isatuximab may be better tolerated..” Is this true? Dara is very well tolerated. Mechanisms for how isa could be more potent than dara are presented, but I don’t understand how there would be less toxicity, and I did not find note of this in the Martin et al reference.

Response: The Martin study states that isatuximab is “well-tolerated” but since direct comparisons with other therapies, including daratumumab-based, are unavailable, the sentence in question has been deleted.

It is not mentioned that the dramatic success of dara in RRMM has lead to the question of whether it should be included as up front therapy, or in patients with smoldering MM. A tantalizing possibility that even deeper responses could be obtained up front.

Response: Language has been added to the daratumumab paragraph (lines 96-102) describing the results of recent studies of daratumumab in recently diagnosed MM, as well as in high-risk smoldering MM. Also see lines 616-618 in the Summary.

Line 133: “..a total of 45 myeloma-based elotuzumab studies, either completed or in progress.” Clearly this and other drugs are coming to the clinic, but with little guidance on their proper role. Dimopoulos et al, New Engl J Med, 2018 just published one of these studies, which added no useful information. Missing is discussion on whether a given agent provides survival benefit over other treatments.

Reference has been added to the recent (2018) NEJM paper by Dimopoulis and mention has been made of the PFS and ORR data reported in that study (lines 146-148) . In addition, language has been added to the elotuzumab paragraph (lines 149-159) discussing the results of three network meta-analytic studies of trials including elotuzumab in RRMM. Also see lines 619-621 in the Summary.

Line 276: "IL-6." Many targets might be combined into a section titled something like, “Failed antibody targets.” E.g. PD-1, PD-L1, DKK-1, CD137, IGF1R, and Sclerostin. How many of the trials on the 2.5 page long Table 4 are failed or aborted? 

Response: The organization of this manuscript has been carefully considered with regard to the placement of individual targets and mAbs in order to make the document as cohesive as possible. For example, DKK1 and sclerostin have been included in the discussion on mAbs intended to prevent bone loss in MM; this placement is deemed more appropriate than inclusion in a separate section devoted to failed or aborted trials. With regard to the placement of PD-1 and PD-L1 blockers, it was considered best to include them in the discussion of checkpoint inhibitors since other members of this class, including CTLA4, KIR, and CD47 blockers remain under consideration. However, Table 4 has been reduced considerably to remove trials that have been placed on clinical holds, although they are noted in the text.  After further review of the document’s organization, the sections on CD33, CD74, NCAM (CD56), and syndecan-1 (CD138) have been moved to a new section titled “8. Additional MAbs”. This move does not disrupt the overall flow of the narrative.

Line 580-581: “…has shown promise for MM in preclinical studies.” There is tremendous excitement about these agents. See https://ash.confex.com/ash/2018/webprogram/Paper109769.htm

Response: The author thanks the reviewer for pointing out the paper to be presented at ASH 2018. A footnote (see Footnote 9 – line 553 - on p. 17) has been added referring to that paper.

Line 622-623: “…the future looks bright for other immunological agents to follow in their wake as MM is transformed from an incurable disease to one more resembling a manageable chronic illness.” This transformation has already happened. For most patients diagnosed in 2018, MM is a chronic disease. The primary issue now is the difficult clinical trial work needed to deliver “the cure” that many in the field believe is on the horizon. We have an embarrassment of riches in the myeloma field regarding antibody therapies, and something of a Wild West approach at present in the clinic. There is little consensus on how to use these agents currently, and some direction would be appreciated.

Response: A portion of the Summary has been re-written. See lines 646-653.

Minor points

Line 31-32: “non-functional paraproteins” Generally this characterization is true of course, but it ignores a) evidence that monoclonal immunoglobulins may target common infectious agents, i.e. Bosseboeuf et al, JCI Insight 2017, and b) the toxic nature of some light chains, e.g. with the development of amyloidosis.

Response: The author thanks the reviewer for pointing out evidence that the paraproteins identified with MM may in fact have certain functions. The word “non-functional” has been deleted and the association of light chains with amyloidosis has been added. See lines 31-34.

Reviewer 3 Report

Monoclonal antibodies are a promising class of therapeutic agent to add to the armamentarium for treating MM. Two mAbs have recently received FDA approval for treating MM and a numerous others are undergoing clinical evaluation. To my knowledge, this article provides for the first time a comprehesive review of all monoclonal antibodies under clinical development in MM. While a useful resource, the breadth of this review can be a little overwhelming. I would suggest that perhaps the monoclonal antibodies in which trials have been discontinued could perhaps just be briefly mentioned in a paragraph together. 

Author Response

I would suggest that perhaps the monoclonal antibodies in which trials have been discontinued could perhaps just be briefly mentioned in a paragraph together.

Response: The author thanks the reviewer for taking the time to read and comment on the manuscript. The organization of this manuscript has been carefully considered with regard to the placement of individual targets and mAbs in order to make the document as cohesive as possible. For example, DKK1 and sclerostin have been included in the discussion on mAbs intended to prevent bone loss in MM; this placement is deemed more appropriate than inclusion in a separate section devoted to failed or aborted trials. With regard to the placement of PD-1 and PD-L1 blockers, it was considered best to include them in the discussion of checkpoint inhibitors since other members of this class, including CTLA4, KIR, and CD47 blockers remain under consideration. However, Table 4 has been reduced considerably to remove trials that have been placed on clinical holds, although they are noted in the text.  After further review of the document’s organization, the sections on CD33, CD74, NCAM (CD56), and syndecan-1 (CD138) have been moved to a new section titled “8. Additional MAbs”. This move does not disrupt the overall flow of the narrative.